

# A comparison of common metrics used to quantify the effectiveness of conservation interventions

Igor Khorozyan

Department of Conservation Biology, Georg-August Universität Göttingen, Göttingen, Germany

## ABSTRACT

**Background:** Evidence-based conservation is urgently needed to identify, apply and promote effective interventions for mitigation of threats and recovery of the natural environment. Estimation of intervention effectiveness is subject to robust study design and statistical analysis, and much progress is documented in these fields. In contrast, little is understood about the accuracy and biases (underestimation and overestimation) of different effectiveness metrics and how they are affected by sample size.

**Methods:** In this study, a dataset ($n$ = 500 cases) consisting of random, positive, integer numbers was simulated to produce frequency input data for the 2 × 2 contingency table. For each case, three metrics of the relative risk, odds ratio and the magnitude of change were calculated, their disparity was estimated and the characteristics of treatment (with intervention) and control (without intervention) samples significantly affecting this disparity were studied by means of linear regression. Further, four case studies from different conservation interventions are provided to support the results.

**Results:** The study has shown that the relative risk and the magnitude of change produce identical estimates of intervention effectiveness only when treatment and control samples are equal, or when the number of target outcomes (e.g., number of livestock killed by predators) in treatment sample reaches zero. In other situations, the magnitude of change gives overestimates or underestimates, depending on relationships between treatment and control sample sizes. The table summarizing the conditions of equalities and biases between these two metrics is provided. These conditions are valid for both reduction-aimed interventions reducing negative target outcomes (e.g., livestock protection to reduce livestock losses to predators) and for addition-aimed interventions increasing positive target outcomes (e.g., establishment of protected areas to increase species presence). No significant effects on the odds ratio were found.

**Conclusion:** Researchers should set equal treatment and control sample sizes so that to produce identical estimates of intervention effectiveness by the relative risk and the magnitude of change. Otherwise, these estimates are biased if produced by the magnitude of change and the relative risk should be used instead. As setting equal treatment and control samples can be impractical, I encourage researchers and practitioners to use the relative risk in estimation of intervention effectiveness. This will not take additional efforts as both metrics are calculated from the same contingency table. Treatment and control sample sizes, along with their sub-samples

Corresponding author
Igor Khorozyan,
igor.khorozyan@biologie.uni-goettingen.de

affected and not by an intervention, should be explicitly reported by researchers to allow independent evaluation of intervention effectiveness. This approach can help obtain more accurate information on intervention effectiveness for making better decisions in conservation actions.

# INTRODUCTION

Coping with globally widespread human pressures on the environment demands the urgent promotion of evidence-based conservation, which is the use of best available evidence to foster informed decision-making in practical conservation (*Pullin et al., 2004*; *Sutherland et al., 2004*; *Sutherland & Wordley, 2017*; *Altringham, Berthinussen & Wordley, 2020*). This concept ensues from, and follows the principles of, the effectiveness revolution which began in the 1970s in medicine (*Pullin & Knight, 2001*). In other words, evidence-based conservation strives to collect, analyze, interpret, synthesize and distribute information on the effectiveness of interventions applied to mitigate threats and improve the status of (semi-)natural ecosystems and their components (*Westgate et al., 2018*; *Altringham, Berthinussen & Wordley, 2020*). Evidence is subject to rigorous scientific assessments and it should be widely published to stimulate replications and reduce the research-implementation gap (*Toomey, Knight & Barlow, 2017*; *Burivalova et al., 2019*). Comparisons of several interventions in the same setting are very useful to select the most effective interventions for further applications (*Smith et al., 2014*; *Schmidt et al., 2019*). Active development of open-access web resources (e.g., Conservation Evidence— www.conservationevidence.com and the Collaboration for Environmental Evidence— www.environmentalevidence.org), professional meetings, analytical tools and outreach awareness-raising is aimed at the wider use of evidence in conservation actions (*Sutherland & Wordley, 2017*; *Westgate et al., 2018*; *Sutherland et al., 2019*; *Pullin et al., 2020*).

Deeper understanding of the importance of intervention effectiveness leads to more scrutiny over the issues of study design and statistical analysis. It is generally agreed that experiments must be controlled and case-control studies, such as comparisons before and after the intervention and comparisons with and without the intervention, are less reliable because they do not have properly matched intervention-free controls to compare with (*Burivalova et al., 2019*; *Christie et al., 2019*). Arguably, the most reliable study designs are before-after-control-impact (BACI), control trials and cross-over, especially when study objects are randomized and experimental procedures are blinded to minimize biases (*Adams, Barnes & Pressey, 2019*; *Christie et al., 2019*; *Treves et al., 2019*). A number of metrics have been used to quantify the effectiveness of interventions, including the relative risk (*Eklund et al., 2017*; *Khorozyan & Waltert, 2019a*; *Khorozyan & Waltert, 2019b*; *Bruns, Waltert & Khorozyan, 2020*; *Khorozyan et al., 2020*), odds ratio (*Knarrum et al., 2006*;

Woodroffe et al., 2007; Wielgus & Peebles, 2014), magnitude of change (Jones & Schmitz, 2009; Green et al., 2013; Miller et al., 2016) and Hedges' d (Van Eeden et al., 2017).

Despite progress in developing the methodology of evidence-based conservation, very little understanding has been obtained regarding differences in effectiveness estimates produced by different metrics. Scientists usually use a single metric per study and it remains unclear whether their estimates are sufficiently accurate or produce biases (underestimation and overestimation) for a variety of reasons. The relative risk and the odds ratio are very similar when events are rare (Stare & Maucort-Boulch, 2016), for example, when predators kill livestock and the effectiveness of livestock protection interventions is evaluated (Khorozyan & Waltert, 2019a; Bruns, Waltert & Khorozyan, 2020). Van Eeden et al. (2018) compiled three independent meta-analyses, each of which used a different metric (relative risk, magnitude of change and Hedges' d), to assemble current information on the effectiveness of livestock protection interventions against predators, but did not compare these metrics from the same case studies.

It is of particular practical importance and scientific interest to compare the metrics calculated from the same standardized matrix produced from the same study. Among the most common matrices is the 2 × 2 contingency table which summarizes the distribution of dichotomous outcomes, such as species A present and species A absent, across the treatment (with intervention) and control (without intervention) samples (Stare & Maucort-Boulch, 2016). The contingency table approach can be used to calculate the relative risk, odds ratio and the magnitude of change with no need for additional information. The effects of treatment and control samples and their relationships on metrics are key to identify possibilities for accurate and unbiased estimation of effectiveness. Larger samples improve the precision of effectiveness estimates (Smith et al., 2014; Christie et al., 2019), but this is not sufficient and the relationships between treatment and control samples may uncover new effects worthy of further investigations. As a researcher is responsible for setting treatment and control sample sizes before the study, this knowledge may provide useful rules-of-thumb for study design, implementation and data analysis.

In this study, I used a simulated dataset to calculate the relative risk, odds ratio and magnitude of change from the 2 × 2 contingency table, estimate their disparity, and find sample characteristics significantly affecting this disparity. I did not consider Hedges' d as this metric requires additional quantitative information (mean and standard deviation) from treatment and control samples, which is not available from contingency tables (Nakagawa & Cuthill, 2007). Four case studies from different conservation interventions are provided and described in order to illustrate and support the results.

## MATERIALS AND METHODS

### Framework

The 2 × 2 contingency table of the numbers of target and alternative outcomes in treatment and control samples was a basis of this study (Fig. 1). The target outcomes are the outcomes that interventions strive to change, otherwise they are designated as the alternative outcomes. As interventions aim to reduce negative target outcomes or add

| | Outcomes | | |
|---|---|---|---|
| | No. target outcomes | No. alternative outcomes | |
| **Treatment = Intervention present** | $N_{t1}$ | $N_{t2}$ | $N_{t1} + N_{t2} = N_t$, treatment sample size |
| **Control = Intervention absent** | $N_{c1}$ | $N_{c2}$ | $N_{c1} + N_{c2} = N_c$, control sample size |
| | $N_{t1} + N_{c1} = N_1$, total number of target outcomes | $N_{t2} + N_{c2} = N_2$, total number of alternative outcomes | |

(left side vertical label: **Intervention**)

**Figure 1  The 2 × 2 contingency table used for estimation of intervention effectiveness.**

positive target outcomes, they were provisionally named as reduction-aimed interventions and addition-aimed interventions. An example of reduction-aimed intervention is livestock protection that aims to reduce or eliminate livestock losses to predators (*Eklund et al., 2017*) and an example of addition-aimed intervention is the establishment of protected areas to increase species presence (*Da Silva et al., 2018*). In these two examples, the target outcomes are the number of lost livestock and the number of sampling units (e.g., grids or transects) with species presence. The alternative outcomes are the number of surviving livestock and the number of sampling units with species absence. The treatment samples are those affected by interventions and the control samples, also known as counterfactuals, are not affected.

The purpose of data analysis from the contingency table is to determine the effectiveness of interventions, that is, whether they significantly decrease the target outcomes for reduction-aimed interventions or significantly increase the target outcomes for addition-aimed interventions. In Fig. 1, $N_{t1}$ stands for the number of target outcomes in treatment sample, $N_{t2}$ for the number of alternative outcomes in treatment sample, $N_{c1}$ for the number of target outcomes in control sample and $N_{c2}$ for the number of alternative outcomes in control sample. The treatment sample size is $N_t = N_{t1} + N_{t2}$ and the control sample size is $N_c = N_{c1} + N_{c2}$.

## Data

I simulated a dataset of $N_{t1}$, $N_{t2}$, $N_{c1}$ and $N_{c2}$ which consisted of random, positive, integer numbers and included 500 independent cases (Data S1). For this, I used the RANDBETWEEN(1, 1,000) function in MS Excel to select random integers between 1 and 1,000. The numbers $N_t$ and $N_c$ were calculated accordingly. It was assumed that each case represents an individual and independent study.

Intervention effectiveness was estimated for each case in comparable metrics of the relative risk (RR%), odds ratio (OR%) and the magnitude of change (D%), all presented in percentages of outcome change (decrease, no change, increase). Equations (1)–(3) use the notations described above. Abbreviations RR, OR and D stand for proportions not transformed to percentages.
$$\text{RR\%} = (\text{RR} - 1) \times 100 = \left(\frac{N_{t1}/N_t}{N_{c1}/N_c} - 1\right) \times 100 \tag{1}$$

$$\text{OR\%} = (\text{OR} - 1) \times 100 = \left(\frac{N_{t1}/N_{t2}}{N_{c1}/N_{c2}} - 1\right) \times 100 \tag{2}$$

$$D\% = \left(\frac{N_{t1}}{N_{c1}} - 1\right) \times 100 \tag{3}$$

The term (RR−1) was taken instead of (1−RR) (*Khorozyan & Waltert, 2019b*; *Bruns, Waltert & Khorozyan, 2020*) to make RR%, OR% and $D\%$ move in the same direction: their negative values indicate a decrease in target outcome, 0 indicate no change and their positive values indicate an increase in target outcome. For the same reason, we inverted the nominator provided in the original equation of $D\%$ (*Jones & Schmitz, 2009*; *Green et al., 2013*; *Miller et al., 2016*). The minimum values of RR%, OR% and $D\%$ are −100%, meaning that the target outcome decreases by 100% when $N_{t1} = 0$.

The measures of disparity between RR%, OR% and $D\%$ were the absolute differences |RR% − $D\%$|, |RR% − OR%| and |OR% − $D\%$|, which I called the disparity distance. The disparity distance can be caused by both underestimation and overestimation by any of these metrics, therefore it was used as an absolute value for standardization.

## Data analysis

As the three effectiveness metrics depend on outcome proportions, the effects of the following outcome ratios on the disparity distance were measured: $N_{t1}/N_{t2}$, $N_{c1}/N_{c2}$, $N_{t1}/N_{c1}$, $N_{t2}/N_{c2}$, $N_{t1}/N_t$, $N_{t2}/N_t$, $N_{c1}/N_c$, $N_{c2}/N_c$ and $N_t/N_c$. The relationships between each of these ratios and the disparity distance were checked for statistical significance and the effect size (coefficient of determination $r^2$) using scatterplots and simple linear regression (*Guthery & Bingham, 2007*; *McKillup, 2012*). Outliers were defined as the points with Cook's distance >1 and removed for further re-analysis. Precision was measured as the 95% confidence intervals of the line and the non-zero slope $\beta_1$, and accuracy as the root mean square error (RMSE) which was normalized (nRMSE) by taking its percentage from the range $y_{max} - y_{min}$ (*Wan et al., 2010*). The effects of outcome ratios on the disparity distance were checked by significance of the $F_{1,n-2}$ statistic at $p < 0.05$ (*McKillup, 2012*).

After significant predictors were found among the outcome ratios, the regression lines of these predictors were built against RR%, OR% and $D\%$ to visualize the disparity distance, with further estimation of precision, accuracy and effect size as described above. The significance of difference between the slopes of RR%, OR% and $D\%$ lines was estimated using the interaction terms in ANCOVA (*McKillup, 2012*). To determine underestimation and overestimation between the metrics, significant outcome ratios were split into two almost equal groups, <1 and >1, and the pairs RR% vs. OR%, OR% vs. $D\%$ and RR% vs. $D\%$ were compared between these two groups by Mann–Whitney test. Chi-square ($\chi^2$) test was applied to compare the frequencies of RR%, OR% and $D\%$. The program IBM SPSS 26.0 was used for statistical analysis throughout the study.

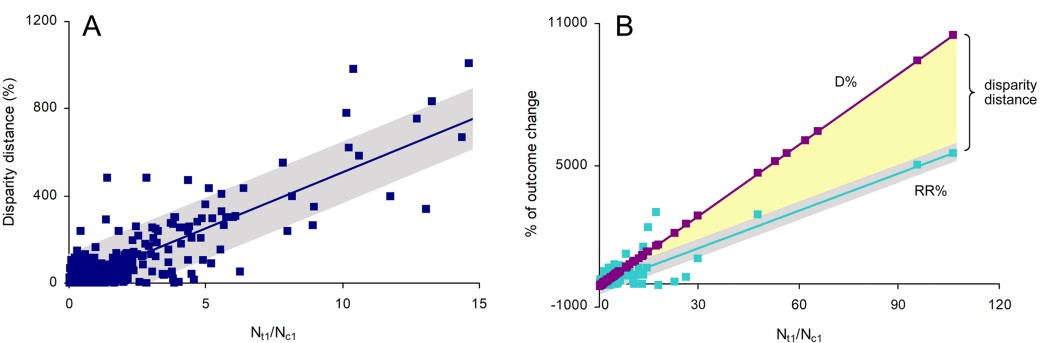

**Figure 2 The effects of the ratio of the number of target outcomes in treatment sample ($N_{t1}/N_{c1}$) on (A) disparity distance between the relative risk RR% and the magnitude of change D% and on (B) RR% and D% separately.** The line on (A) and the RR% line on (B) are indicated with their 95% confidence intervals.

## Case studies

To demonstrate wide applicability of study results, I attempted to use four case studies focused on mammals, birds, reptiles and landscape restoration, dealing with reduction-aimed and addition-aimed interventions. I used my library (*Knarrum et al., 2006*), Conservation Evidence (*Newell, 2019*), and Google Scholar (http://scholar.google.com, search words "turtle 2 × 2 contingency table", *Sampaio, 2018*; "restoration 2 × 2 contingency table", *Andruk, 2014*) to find the literature. These publications contain data produced from controlled experiments. Original data of these studies were used to produce contingency tables and calculate the metrics of RR%, OR% and D% for comparison.

## RESULTS

Disparity distances between OR% and the other two metrics were significantly affected only by the odds of target outcomes in treatment sample ($N_{t1}/N_{t2}$), which is natural as OR% is calculated from these odds (see Eq. (2) above). Therefore, only RR% and D% were considered further in this study. The only outcome ratio significantly affecting the disparity distance between RR% and D% was the ratio of the number of target outcomes in treatment sample ($N_{t1}$) to that in control sample ($N_{c1}$), that is, $N_{t1}/N_{c1}$ (Fig. 2). This effect was significant ($F_{1,480} = 1{,}194.49$, $p < 0.001$), precise and positive ($\beta_1 = 51.402 \pm 1.487$, 95% CI [48.480–54.324]) and strong ($r^2 = 0.713$), but not accurate (nRMSE = 575.63%). As D% is calculated from $N_{t1}/N_{c1}$, its line was statistically perfect ($r^2 = 1$). The effect of $N_{t1}/N_{c1}$ on RR% was also significant ($F_{1,489} = 2{,}081.50$), precise and positive ($\beta_1 = 52.571 \pm 1.152$, 95% CI [50.307–54.835]), strong ($r^2 = 0.810$) and accurate (nRMSE = 3.33%) (Fig. 2). The slopes of D% and RR%, which embraced the disparity distance within, were statistically different (ANCOVA: $F_{1,982} = 2{,}036.80$, $p < 0.001$).

When $N_{t1}/N_{c1}$ was less than 1, there was a significantly higher number of cases with D% < RR% ($n = 179$) than with D% > RR% ($n = 75$), and this difference was significant ($\chi^2 = 42.583$, $p < 0.001$). The estimate D% was significantly lower than RR% (Mann–Whitney U = 25,913.5, $p < 0.001$). These inequalities depended on the ratio of

**Table 1 The scheme of disparities between the magnitude of change (D%) and the relative risk (RR%) in estimating the effectiveness of conservation interventions depending on relationships between the numbers of target outcomes and sample sizes.** See the contingency table in Fig. 1 for more details.

| Numbers of target outcomes[1] | Treatment and control sample sizes[2] | Relationships between D% and RR%[3] | Explanation |
|---|---|---|---|
| $N_{t1} < N_{c1}$ | $N_t > N_c$ | $\lvert D\%\rvert < \lvert RR\%\rvert$ | Underestimation by D% |
| | $N_t < N_c$ | $\lvert D\%\rvert > \lvert RR\%\rvert$ | Overestimation by D% |
| | $N_t = N_c$ | D% = RR% | Same estimate by both metrics |
| $N_{t1} > N_{c1}$ | $N_t > N_c$ | $\lvert D\%\rvert > \lvert RR\%\rvert$ | Overestimation by D% |
| | $N_t < N_c$ | $\lvert D\%\rvert < \lvert RR\%\rvert$ | Underestimation by D% |
| | $N_t = N_c$ | D% = RR% | Same estimate by both metrics |
| $N_{t1} = N_{c1}$ | $N_t > N_c$ | D% = 0, D% > RR% | No effect by D%, decrease of target outcome by RR% |
| | $N_t < N_c$ | D% = 0, D% < RR% | No effect by D%, increase of target income by RR% |
| | $N_t = N_c$ | D% = RR% = 0 | No effect of intervention |
| $N_{t1} = 0$ | – | D% = RR% = −100% | Maximum reduction of target outcome by 100% |

Notes:
[1] $N_{t1}$, number of target outcomes in treatment sample; $N_{c1}$, number of target outcomes in control sample.
[2] $N_t$, treatment sample size and $N_c$, control sample size.
[3] The absolute estimates are provided because D% and RR% can be both positive and negative.

treatment sample size ($N_t$) to control sample size ($N_c$): $\lvert D\%\rvert < \lvert RR\%\rvert$ when $N_t/N_c$ was higher than 1 and $\lvert D\%\rvert > \lvert RR\%\rvert$ when $N_t/N_c$ was less than 1. When $N_t/N_c$ is equal to 1, D% = RR%. The absolute estimates are provided because D% and RR% can be both positive and negative. The general scheme of underestimations, overestimations and equalities of D% and RR% is shown in Table 1.

When $N_{t1}/N_{c1}$ was higher than 1, there was a significantly higher number of cases with D% > RR% ($n = 172$) than with D% < RR% ($n = 64$), and this difference was significant ($\chi^2 = 49.424$, $p < 0.001$). The estimate D% was significantly higher than RR% (Mann–Whitney $U = 18,779.0$, $p < 0.001$). Like in the previous scenario, these relationships depended on $N_t/N_c$: $\lvert D\%\rvert > \lvert RR\%\rvert$ when $N_t/N_c$ was higher than 1 and $\lvert D\%\rvert < \lvert RR\%\rvert$ when $N_t/N_c$ was less than 1. Again, when $N_t/N_c$ is equal to 1, D% = RR% (Table 1).

When $N_{t1}/N_{c1}$ is equal to 1, D% indicates no effect of interventions regardless of relationships between $N_t$ and $N_c$ whereas RR% indicates a decrease or increase of target outcomes depending on sample sizes. When $N_t = N_c$, both D% and RR% estimate no effect of interventions (Table 1).

## Case studies

### Reduction-aimed interventions. Case 1: Use of bell collars to deter brown bears (Ursus arctos) from ewes in Norway

Bell collars are often used to easily locate grazing livestock. However, they also may serve as acoustic deterrents from predators, thus reducing livestock mortality from predator attacks and relieving tensions between local people and conservationists. To test this,

**Table 2 The numbers of ewes with and without bell collars killed by brown bears (*Ursus arctos*) in Norway in 1994 (*Knarrum et al., 2006*).**

|  | No. killed | No. not killed | Total |
|---|---|---|---|
| With bells | 10 | 9 | 19 |
| Without bells | 7 | 27 | 34 |
| Total | 17 | 36 | 53 |

**Table 3 The numbers of green turtle (*Chelonia mydas*) nests with and without metal nets predated by Nile monitors (*Varanus niloticus*) in Guinea-Bissau in 2016 (*Sampaio, 2018*).**

|  | No. predated | No. not predated | Total |
|---|---|---|---|
| With metal nets | 3 | 23 | 26 |
| Without metal nets | 19 | 45 | 64 |
| Total | 22 | 68 | 90 |

*Knarrum et al. (2006)* compared depredation rates among the ewes with and without bell collars caused by brown bear attacks. The contingency table is shown in Table 2. In this sample, $N_{t1} > N_{c1}$ and $N_t < N_c$, therefore the absolute estimate of $D$% should be lower than RR% (Table 1). Our results show that bell collars increased ewe mortality from bear attacks by $D$% = 42.8% and RR% = 155.6% instead of decreasing it, possibly because bears learned to associate the sound of bells with food. So, bell collars were ineffective and even counter-productive in protecting ewes from bears.

### Reduction-aimed interventions. Case 2: Use of metal nets to protect green turtle (Chelonia mydas) nests from predation in Guinea–Bissau

The green turtle is globally endangered due to uncontrolled harvesting of animals and eggs, and the main nesting grounds in undisturbed sites need protection from predators. In this study, randomly selected turtle nests were covered by 1 m² metal nets to prevent access to Nile monitors (*Varanus niloticus*) and compared with open control nests in terms of monitor predation (*Sampaio, 2018*; Table 3). As $N_{t1} < N_{c1}$ and $N_t < N_c$ in this sample, the absolute $D$% is to be higher than RR% (Table 1). The result is $D$% = −84.2% and RR% = −61.1%, confirming that metal nets were effective in reducing nest predation.

### Addition-aimed interventions. Case 3: Use of nest forms to increase the occupancy of nest boxes by common swifts (Apus apus) in the UK

As swifts are endangered in the UK, principally due to the lack of suitable nesting sites, nest boxes are set up to recover the population. Yet, the optimal design of nest boxes is insufficiently known and placing a nest form inside nest boxes is a promising method to increase the occupancy of nest boxes by swifts. In this study, the numbers of occupied nest boxes were compared between the boxes with and without nesting forms (*Newell, 2019*; Table 4). Here, $N_{t1} > N_{c1}$ and $N_t < N_c$, meaning the absolute $D$% being lower than RR% (Table 1). Calculations show that nesting forms increased the occupancy of nesting

**Table 4 The numbers of nest boxes with and without nest forms occupied by swifts (*Apus apus*) in the UK in 2009–2018 (*Newell, 2019*).**

|  | No. occupied | No. not occupied | Total |
|---|---|---|---|
| With nest forms | 30 | 32 | 62 |
| Without nest forms | 12 | 68 | 80 |
| Total | 42 | 100 | 142 |

**Table 5 The numbers of hardwood sprouts in plots with and without prescribed fire in the USA in 2009–2012 (*Andruk, 2014*).**

|  | No. sprouts post-treatment | No. sprouts pre-treatment | Total |
|---|---|---|---|
| With fire | 669 | 247 | 916 |
| Without fire | 152 | 92 | 244 |
| Total | 821 | 339 | 1,160 |

boxes by swifts by $D\% = 150.0\%$ and $RR\% = 222.6\%$, and their use is effective for swift conservation.

### Addition-aimed interventions. Case 4: Use of prescribed fire to recover woodland in the USA

Prescribed fire is a common intervention used to recover natural vegetation of arid landscapes. *Andruk (2014)* tested how the fire increased the number of sprouts of hardwood plants such as sugarberry (*Celtis laevigata*), black cherry (*Prunus serotina*) and others in woodland of central Texas. In this study design, the number of sprouts post-treatment served as a target outcome and the number of sprouts pre-treatment was an alternative outcome (Table 5). With $N_{t1} > N_{c1}$ and $N_t > N_c$, the absolute $D\%$ should be higher than $RR\%$ (Table 1). The result is that prescribed fire increased the number of hardwood sprouts by $D\% = 340.1\%$ and $RR\% = 17.2\%$.

## DISCUSSION

This study has shown that the relative risk and the magnitude of change produce identical estimates of intervention effectiveness only when treatment and control samples are equal, or when the number of target outcomes in treatment sample is zero (Table 1). In other situations, the magnitude of change gives overestimates or underestimates, depending on relationships between treatment and control sample sizes. These patterns were demonstrated by means of theoretical simulations and practical case studies, for both reduction-aimed and addition-aimed conservation interventions. Overestimation takes place when the relationships between treatment and control sample sizes, and between the numbers of target outcomes in treatment and control samples, are in the same direction— that is, both are lower, or both are higher, in treatment than in control samples (Table 1). When these relationships go in opposite directions, the magnitude of change gives underestimation of effectiveness. The higher the dissimilarity between treatment and

control sample sizes, the higher the magnitude of overestimation or underestimation, which is best illustrated by case studies 1 and 4 (*Knarrum et al., 2006*; *Andruk, 2014*). Disparity between the two metrics further increases with the increasing ratio of the number of target outcomes in the treatment sample to that in the control sample (Fig. 2).

Information about the accuracy of different effectiveness metrics under variable relationships between treatment and control sample sizes is practically important because the determination of sample size is the researcher's prime task to design a good study (*Smith et al., 2014*; *Christie et al., 2019*). When samples are subject to study design, as in the cases with bell-collared and uncollared ewes (case study 1; *Knarrum et al., 2006*), net-covered and uncovered turtle nests (case study 2; *Sampaio, 2018*) and swift nest boxes with and without nest forms (case study 3; *Newell, 2019*), they should be set to be equal or similar between treatment and control. In contrast, when sample size is beyond control, for example, it represents the number of sprouts in case study 4 (*Andruk, 2014*), it is not possible to secure the equality of treatment and control sample sizes. Some difference between treatment and control sample sizes is still acceptable, as in case study 3, but its increase leads to inaccurate estimation of intervention effectiveness by the magnitude of change. As a result, this study not only supports an appeal for balanced study designs in ecological research (*Fletcher & Underwood, 2002*), but also indicates the conditions when biases in estimation are possible.

The only situation when the relationships between treatment and control samples do not matter is when the number of target outcomes in treatment sample is 0. This is the best outcome for reduction-aimed interventions which is quantified by the relative risk and the magnitude of change as the maximum reduction of target outcome by 100% (Table 1). Interventions applied to reduce, inter alia, the rates of livestock depredation (*Eklund et al., 2017*; *Van Eeden et al., 2018*), deforestation (*Burivalova et al., 2019*) and poaching (*Kragt et al., 2020*) are all aimed at reduction of target outcomes and their ultimate goal is to nullify them. However, such maximum effectiveness cannot be assumed a priori even for interventions which often show excellent results, like electric fences for reduction of livestock losses (*Khorozyan & Waltert, 2019a*, *2019b*), because the effectiveness is site-specific. Therefore, proper setting of treatment and control sample sizes is essential anyway. These sample sizes, as well as the numbers of target and alternative outcomes as their sub-samples (Fig. 1), should be explicitly reported by researchers to allow independent evaluation of intervention effectiveness.

If designed treatment and control samples are unlikely to be similar, it is most reasonable to expect the treatment sample to be smaller than the control sample due to significant efforts and resources invested to the process of treatment (*Eklund et al., 2017*; *Snijders et al., 2019*). This was confirmed by the first three case studies where sample sizes were determined by researchers (*Knarrum et al., 2006*; *Sampaio, 2018*; *Newell, 2019*). In this situation, three scenarios are possible: (1) if the number of target outcomes in treatment sample is lower than that in control sample—overestimation by the magnitude of change; (2) if the number of target outcomes in treatment sample is higher than that in control sample—underestimation by the magnitude of change; and (3) if the numbers of target outcomes in treatment and control samples are equal—the magnitude of

change estimates no effectiveness and the relative risk indicates an increase in target outcome (Table 1).

Why are overestimation and underestimation specified for the magnitude of change in regard to the relative risk, but not vice versa? The point is that the magnitude of change does not consider sample sizes and is thus a less reliable metric. The relative risk is a ratio of the probability of target outcome with the intervention to that probability without the intervention (*Eklund et al., 2017*; *Khorozyan & Waltert, 2019a*), so it represents a balanced metric accounting for treatment and control sample sizes. It is one of the main effectiveness metrics in medicine (*Stare & Maucort-Boulch, 2016*), a discipline which is well advanced in research of evidence-based intervention effectiveness (*Pullin & Knight, 2001*; *Sutherland et al., 2004*; *Sutherland & Wordley, 2017*; *Altringham, Berthinussen & Wordley, 2020*). In contrast, the magnitude of change is based only on the difference between the numbers of target outcomes (*Jones & Schmitz, 2009*; *Green et al., 2013*; *Miller et al., 2016*), yet sample sizes are critically important and their ignorance can drastically change the effectiveness estimates. A vivid example is the situation when the number of target outcomes in treatment sample is equal to that in control sample. In this case, the magnitude of change fails to detect the effectiveness and quantifies it as 0, whereas the relative risk estimates the effectiveness to increase or decrease depending on sample sizes (Table 1). In extreme cases of imbalanced samples, with high dissimilarity between treatment and control samples, the magnitude of change can even show an opposite direction of effectiveness in comparison with the relative risk. For example, imagine a fictitious case when the numbers in Table 2 are changed as follows: number of killed ewes with bells—1, number of surviving ewes with bells—7, number of killed ewes without bells—3 and number of surviving ewes without bells—97. So, the treatment sample size is 8 and the control sample size is 100. In this case, the magnitude of change shows a reduction of depredation rate by 66.7%, but the relative risk shows an increase by 316.7%.

I hope that this study will stimulate more active implementation of well-designed scientific and practical research in evidence-based conservation so that to make tangible and long-lasting positive changes in the state of the environment.

## CONCLUSIONS

Researchers should set equal treatment and control sample sizes in order to produce identical estimates of intervention effectiveness by the relative risk and the magnitude of change. When treatment and control samples are unequal, these estimates are biased if produced by the magnitude of change and the relative risk should be used instead. As setting equal treatment and control samples can be problematic for practical reasons, I encourage researchers and practitioners to use the relative risk in estimation of intervention effectiveness. This will not take additional efforts as both metrics are calculated from the same contingency table. Researchers should report treatment and control sample sizes, and the numbers of target and alternative outcomes within these samples, to let the readers evaluate intervention effectiveness independently. Overall, this approach can strengthen study design and data analysis by scientists, increase the

reliability of results for practitioners, and facilitate decision-making by conservation managers based on accurate information.

## ACKNOWLEDGEMENTS

I thank A. Christie and an anonymous reviewer for providing insightful comments, which significantly improved the quality of the manuscript.

### Funding

The author received no funding for this work.

### Competing Interests

The author declares that he has no competing interests.

### Author Contributions

- Igor Khorozyan conceived and designed the experiments, performed the experiments, analyzed the data, prepared figures and/or tables, authored or reviewed drafts of the paper, and approved the final draft.

### Data Availability

Raw dataset and SPSS synthax are available in the Supplemental Files.

### Supplemental Information

Supplemental information for this article can be found online at http://dx.doi.org/10.7717/peerj.9873#supplemental-information.

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
