# Peer review of "A comparison of common metrics used to quantify the effectiveness of conservation interventions"

_PeerJ, doi:10.7717/peerj.9873_

## Round 0.1 · original submission · Minor Revisions

I received comments from two reviewers on your manuscript. Your paper should become acceptable for publication pending suitable minor revision and modification of the article in light of the reviewers' comments. Please, follow ALL suggestions and send me your revised manuscript as soon as you can.

·

Basic reporting

The manuscript is generally well written and provides a nice overview and background of evidence-based conservation, and the need for meta-research such as this to understand the best ways to reliably quantify the effectiveness of conservation interventions.

Other comments:
Line 90/91: The Conservation Evidence database can be cited directly. Sutherland, W. J., Taylor, N. G., MacFarlane, D., Amano, T., Christie, A. P., Dicks, L. V., ... & Petrovan, S. O. (2019). Building a tool to overcome barriers in research-implementation spaces: The Conservation Evidence database. Biological Conservation, 238, 108199.

A more general point is that more literature could be cited about ‘unbalanced designs’ in ecology – there is a lot of discussion around this in reference to BACI-type designs. This imbalance is related to different numbers of treatment and control samples. e.g. Fletcher, D. J., & Underwood, A. J. (2002). How to cope with negative estimates of components of variance in ecological field studies. Journal of Experimental Marine Biology and Ecology, 273(1), 89-95.

And some minor grammatical/spelling issues:

L113-114 This sentence just doesn’t make sense – possibly some words have been deleted?
L122 – replace ‘like’ (a bit colloquial) with ‘such as’
L127 – larger samples do not always improve the accuracy of effectiveness estimates, they improve their precision. Greater accuracy comes from removing biases.
L272 – is ‘acoustical’ the right word, is it not ‘acoustic’.
L326 – ‘The higher is the’ should be ‘The higher the’.
L327 – ‘the higher is the’ should be ‘the higher the’.
L327 – ‘magnitude’ instead of ‘scope’
L330 – should be ‘in the treatment sample compared to that in the control sample’
L350 – ‘a priori’ needs to be italicised
L352 – no need to say ‘it can be different in other sites’ as you’ve said ‘site-specific’.
L364 and conclusion – surely the advice should be to use standardised effect sizes regardless as they account for sample size? Yet you said that is not the aim of this study. Given in most cases the number of treatment and control samples are unlikely to be the same and it may be difficult to always ensure that these numbers are the same. The RR% and OR% do not take much extra effort to calculate given that all the data needed to calculate them should be available whether or not you are calculating the effectiveness using D%. Is it because you feel that D% is a more intuitive metric for practitioners to communicate? I feel it’s a little strange to show how biased the D% estimator can be and then fail to discourage its use? I can just see it’s easier for practitioners testing interventions to use the OR or RR and not be constrained by making the number of treatment and control samples the same.

Experimental design

Thankfully I was able to replicate the main results of the simulations in this paper in R with relative ease based on the methods described, but it would be nice if the author supplied code (either for R or SPSS) so that others could do the same.

Nevertheless I do have a few questions (these may be because I wasn’t able to exactly replicate some of the finer details of the analyses, or my errors, but should be checked):
I found significant effects on disparity distance |RR%-D%| for the following ratios: nt1/nc1 (as reported), nt1/nt, nc1/nc, nt2/nt, nc2/nc, and nt/nc. That made me concerned about L237-239.

I also found significant effects on disparity distance |RR%-OR%| and |RR%-D%|. This left me concerned about L236-237.

If code had been supplied to replicate these analyses it would have been a lot easier to check for possible errors and explicitly show what choices were made in analysis. Overall, I am not too concerned about these issues as I was able to replicate the main findings, which are the focus of the paper and make logical sense. I am just wondering about what these other possible comparisons might show and whether they are interesting to include in the manuscript. If code can be provided to show these are errors in replication then that would be ideal, and will be appreciated by readers who may want to go to conduct further research into this issue.

I must also praise the author on finding practical, real-world case studies to show how the simulation results apply in the real-world – that also gives the reader a lot more confidence in these findings and helps to drive home the point that D% is likely to under/overestimate RR% in different situations.

I think the research questions are well defined and the knowledge gap this fills is clear.

Validity of the findings

I must admit I am a bit surprised that these findings are not known already, given that they are relatively fundamental since RRs and ORs form the basis of so much of evidence-based medicine. I imagine these findings are illustrated or at least alluded to in books on meta-analysis (e.g. Koricheva, J., Gurevitch, J., & Mengersen, K. (Eds.). (2013). Handbook of meta-analysis in ecology and evolution. Princeton University Press.).

Nevertheless, while these results aren’t necessarily that surprising to those working in quantitative fields of evidence synthesis and meta-analysis, this doesn’t mean that this paper isn’t needed to reinforce the point that many conservationists and practitioners who conduct tests of interventions are unaware or poorly understand these basic statistical concepts and issues. This message is important and needs to be heard by those testing interventions and reporting metrics of effectiveness for conservation studies – if studies are not designed properly, or with due consideration to the metric used, then their reported results will be misleading to readers, or hard for those in evidence synthesis to convert into standardised effect sizes (e.g. ORs and RRs). Accurate reporting of sample sizes would help a lot and could be mentioned in the discussion. I also think the paper also illustrates the point that evidence-based conservation is still playing catch-up with evidence-based medicine – maybe more could be made of that point in the discussion.

However, as I said previously, I feel the fact that in the conclusions you do not discourage the use of D%, or at least the presentation of OR% and RR% alongside it in all studies, a little strange given your findings. I think the discussion and conclusion can be improved upon prior to acceptance to iron out these messages.

Reviewer 2 ·

Basic reporting

The article is well structured and is logical and coherent. The central idea (and its wider relevance) is introduced clearly and directly. The study design is then presented and the manuscript flows well from there. The manuscript is well thought out and pitched and is clear and simple to follow.
It uses up-to-date literature from across disciplines and across the globe. The use of references is comprehensive; I did not see any obvious omissions.
The figures and tables are appropriate and well constructed. The raw data is notably accessible and clear.
The use of English is professional throughout. In a small number of cases I spotted the possibility to improve the English expression and I have included these suggestions on the manuscript itself.

Experimental design

The manuscript presents the results of original research that fits the Aims and Scope of the journal well. It is primarily of relevance in Biological and Environmental Science but, in the Introduction, draws context thoughtfully from Medical research.
The study is, in some respects, simple. I use that term here as a positive. The research question is clearcut, relevant and meaningful. The reason for undertaking the study is ultimately to increase conservation outcomes from management interventions. The study aims to do that by ensuring that studies are meaningfully designed and analysed by those wanting to undertake management interventions. The study provides an important way forward in this regard.
The design is rigorous with both a theoretical and a practical component. The analysis is targeted, clear and rigorous. It is easy to follow and the results are understandable to a wide audience. This research will be easy to replicate. The interpretations flow from the results in a straight-forward manner.

Validity of the findings

The underlying data are available and have been put together in a way that is easy to follow. The analysis would be easy to replicate.
The conclusions are tightly linked to the findings of the analysis and the outcomes are clear and straightforward. The Discussion is free of speculation. Overall the work holds together very nicely.

Additional comments

A couple of minor points that I suggest to think about covering in the revision.
First, although Hedges' d is mentioned in the Introduction it doesn't appear to come up again. Please include a brief justification for not considering it further.
Second, some more information as to how the four case studies were chosen will be helpful. It is mentioned that they were chosen at random but how was this achieved?

Annotated reviews are not available for download in order to protect the identity of reviewers who chose to remain anonymous.

---

## Round 0.2 · accepted · Accept

I reviewed the updated version of your manuscript and decided to accept it for publication in PeerJ. Congratulations again!